Acceptance of the German e-mental health portal www.psychenet.de: an online survey

Tlach Lisa
Thiel Juliane
Härter Martin
Liebherz Sarah s.liebherz@uke.de
Dirmaier Jörg
Department of Medical Psychology, University Medical Center Hamburg-Eppendorf, Hamburg, Germany
Leach Liana
Electronic publication date: 2016 Jul 19
Publication date: 2016
Volume: 4
Electronic Location ID: e2093
Received 2015 Dec 15; Accepted 2016 May 9
Copyright: ©2016 Tlach et al.
Copyright year: 2016
Copyright holder: Tlach et al.
License: This is an open access article distributed under the terms of the Creative Commons Attribution License, which permits unrestricted use, distribution, reproduction and adaptation in any medium and for any purpose provided that it is properly attributed. For attribution, the original author(s), title, publication source (PeerJ) and either DOI or URL of the article must be cited.
License URL: https://creativecommons.org/licenses/by/4.0/

Keywords: Health information, Process evaluation, Internet, Mental health

Funding: German Federal Ministry of Education and Research 01KQ1002B This work was supported by the German Federal Ministry of Education and Research (grant number 01KQ1002B). The funders had no role in study design, data collection and analysis, decision to publish, or preparation of the manuscript.

==============================
Background. Taking into account the high prevalence of mental disorders and the multiple barriers to the use of mental health services, new forms of fostering patient information, involvement, and self-management are needed to complement existing mental health services. The study aimed at investigating acceptance regarding design and content of the e-mental health portal www.psychenet.de.

Methods. An online cross-sectional survey was conducted between May 2013 and May 2015 using a self-administered questionnaire including items on perceived ease of use, perceived usefulness, attitude towards using, and perceived trust. Effects of different participants’ characteristics on the portals’ acceptance were analyzed.

Results. The majority of the N = 252 respondents suffered from mental disorders (n = 139) or were relatives from persons with mental disorders (n = 65). The portal was assessed as “good” or “very good” by 71% of the respondents. High levels of agreement (89–96%) were shown for statements on the perceived ease of use, the behavioral intention to use the portal, and the trustworthiness of the portal. Lower levels of agreement were shown for some statements on the perceived usefulness of the portals’ content. There were no effects of different participants’ characteristics on the perceived ease of use, the perceived usefulness, the attitude towards using the website and the perceived trust.

Discussion. This survey provides preliminary evidence that the e-mental health portal www.psychenet.de appears to be a usable, useful and trustworthy information resource for a broad target group. The behavioral usefulness of the portals’ content might be improved by integrating more activating patient decision aids.

Introduction

Over a third of the total EU population suffers from mental disorders with anxiety and mood disorders being the most frequent mental disorders (Wittchen et al., 2011). However, mental disorders are often not detected; only about one third of patients receives adequate treatment, and access to treatment is complicated by system-related barriers (Mack et al., 2014). In most epidemiological studies, service use of mentally ill people ranges between 2% and 18% (Wang et al., 2007). Given the structural problems of the mental health care system, new forms of fostering patient information, involvement, and self-management are needed to complement existing mental health services. Therefore the development of innovative treatment approaches that are available to a large population is recommended (Christensen & Petrie, 2013).

Bridging the gap through web-based health applications

The Internet is widely seen as an effective complementary source for addressing these issues. As it reaches a large number of people, a reduction of barriers to the use of health services is facilitated by anonymity and high accessibility. It holds the opportunity to deliver interactive content that is tailored to the needs of the target group at comparatively low cost to a large number of users at the time, place and learning speed the individual user prefers (Arnberg et al., 2014). Internationally, health services have increasingly expanded into online environments leading to the development of e-mental health services that are designed to complement, rather than replace existing mental health services. They hold the opportunity to reach people who live in remote areas or those with disabilities and without easy access to health care services (Anderson et al., 2013; Benavides-Vaello, Strode & Sheeran, 2013; Carrard et al., 2006). Furthermore, people who refuse to seek out traditional services, especially those who wish to remain anonymous, may utilize e-mental health services (Townsend, Gearing & Polyanskaya, 2012). E-health services may empower patients to participate in treatment choices and to take control and responsibility about their own health and care by improving access to services and information (Alpay et al., 2010; Alpay, Van der Boog & Dumaij, 2011; Xie et al., 2013). A German national survey found that people increasingly take advantage of these opportunities (Eichenberg, Wolters & Brähler, 2013).

However, the quality and usability of mental health information on the World Wide Web is limited (Reavley & Jorm, 2011) and the effectiveness of e-health interventions is limited by high attrition rates (Geraghty et al., 2013); most users visit health intervention websites only once (Brouwer et al., 2010; Verheijden et al., 2007). Additionally, reading levels of web-based patient materials are partially too high for the average user, not taking into account the large variance of health literacy in the population (Stossel et al., 2012). As persons with lower educational levels and respective persons with lower literacy levels might show less beneficial effect by using patient education materials (Goossens et al., 2014; Murphy et al., 2000), effects of educational levels on the acceptance—among other participants’ characteristics—should be accounted for in the interpretation of evaluation results.

The German e-mental health portal

A current project being part of the public-funded intersectoral research network, psychenet—the Hamburg Network for Mental Health is aimed at developing and evaluating an e-mental health portal. With psychenet, the Federal Ministry of Education and Research contributes to strengthening healthcare regions in Germany by establishing new trans-sectoral cooperations and by implementing and evaluating selected health care innovations (Härter et al., 2012). The portal www.psychenet.de is intended to increase the users’ knowledge and to empower them to be active partners in medical decisions and the management of their mental strain.

In a first step, a basic version of the portal (comprising evidence-based patient information on a wide range of mental disorders and information about local treatment services) was developed to complement a region-wide awareness campaign on mental health in the metropolitan area of Hamburg that includes an award-winning media campaign (placards, cinema ads, radio ads) and specific educational projects (Härter et al., 2012). In order to obtain first evidence about the usability of the website, common web metrics were obtained via open source web analysis tools (e.g., Google Analytics). As a following project step, various modules have been developed for six of the most common mental disorders—depression, somatoform disorders, eating disorders, alcohol use disorders, psychotic disorders and anxiety disorders (Wittchen et al., 2011); e.g., patient decision aids (PtDAs), self-help tools, and screening tools. According to the International Patient Decision Aid Standard (IPDAS) collaboration criteria (Elwyn et al., 2006), the development of the modules has been based on a comprehensive mixed-methods needs assessment (focus groups, online-survey) among patients, relatives, and health care professionals. The technical development of the website has been commissioned by a professional web-design agency. The design and content of the portal and the results of the website using web metrics are described in detail elsewhere (Dirmaier et al., 2015). While the information about treatment services refer to the metropolitan area of Hamburg and the media campaign and specific health education projects were restricted to this region, the other tools (e.g., evidence based patient information (including fact sheets on several mental disorders and other basic facts concerning mental health as well as PtDAs) and screening tools) are not targeted specifically to this region.

The present study aimed at investigating acceptance regarding design and content of the basic version of the e-mental health portal www.psychenet.de addressed at individuals with mental disorders, their relatives, service providers, and the interested public. The portal should be assessed through the following aspects of acceptance: (1) perceived ease of use, (2) perceived usefulness, (3) attitude towards using the website, (4) perceived trust, and (5) overall evaluation. A further aim was to explore effects of different participants’ characteristics (sex, age, educational level, place of residence, experience with mental disorders, first time/multiple portal users, participation before or after the integration of the first PtDA) on the portals’ acceptance.

Methods

Design and participants

The research team employed an online cross-sectional study using a self-administered survey. Online convenience sampling was conducted on our e-mental health portal www.psychenet.de. On each page of the portal, teasers were sited linking to a short invitation to participate in the survey. Users being interested were referenced to the survey that was arranged following detailed information about the studies’ aim, procedure, and data security. Adult users (18 years or over) who gave written informed consent to participate (asked at the beginning of the questionnaire) as well as consent to data use (asked when participants had finished the questionnaire) were included in the analyses. There were no additional inclusion or exclusion criteria.

Ethics statement

Approval for the study was obtained from the ethics committee of the Hamburg Medical Association (Process number: PV4157).

Data collection

The data were collected between May 2013 and May 2015 (24 months). A short, face-validated questionnaire comprising 33 items was developed for the study. The questionnaire comprised 3 main sections: (1) baseline characteristics, (2) acceptance and usability, and (3) overall evaluation.

Baseline characteristics

Baseline characteristics were elicited using 4 items on sociodemographic variables (age, gender, education, postal code). Furthermore, 3 items were used to explore respondents’ experience with mental disorders (4 options), how they accessed the website (3 options), and how they learned about the website (8 options including the option for a free answer). Previous internet use was explored on a 3-point scale (“(almost) every day,” “at least once a week,” “at least once a month”) and frequency of use of the portal was elicited on a 4-point scale (“first time,” “<5 times,” “>5 times,” “>10 times”).

Acceptance and usability

In order to assess the acceptance and the usability of the portal, respondents rated up to 22 items on a 4-point Likert scale (1=disagree, 2=somewhat disagree, 3=somewhat agree, 4=agree). Number of scale points and wording of the Likert scale were defined based on Chang (1994). According to a previous study on the acceptance of an e-health application (De Graaf et al., 2013), participants were asked to rate statements covering three dimensions of the Technology Acceptance Model (TAM); see Davis (1989) and Chau & Hu (2002): perceived ease of use (8 items), perceived usefulness (10 items including 2 filter items for respondents being affected by mental disorders and 1 filter item for respondents being a relative of a person with mental disorders), and attitude towards using (2 items). The TAM dimensions were added by the dimension perceived trust (2 items) as it was shown to be a relevant quality criterion as seen by patients with long-term conditions and caregivers (Kerr et al., 2006) and it affects consumers’ acceptance of health technologies (Lemire et al., 2008; Wu et al., 2008).

Overall evaluation

In order to elicit an overall rating of the portal, respondents were asked to rate the portal on a 6-point scale based on the grading system used in German schools (1=very good, 2=good, 3=satisfactory, 4=sufficient, 5=deficient, 6=insufficient). Finally, a facultative open field for comments and suggestions for improvements was provided. Before the questionnaire was used, it was pilot tested among 10 student assistants and research assistants not participating in this study.

Data analysis

Quantitative data analysis

The professional web-based online survey software EFS Survey (Questback GmbH) was used for the electronic data collection. The statistical software package PASW Statistics 18 (SPSS Inc., Chicago IL) was used to analyze the data. Data were primarily evaluated by quantitative descriptive data analysis. In order to quantify responses, means, standard deviations, and frequency distributions were calculated for each item on acceptance. A total score for each dimension (perceived ease of use, perceived usefulness, attitude towards using, perceived trust) was calculated by summing the scale’s single items. Moreover, median, range, and frequency distribution were calculated for the overall rating.

To explore effects of different participants’ characteristics (sex, age, educational level, place of residence, experience with mental disorders, first time/multiple portal users, participation before or after the integration of the first PtDA) on the acceptance and usability of the website, one-way analyses of variance (ANOVAs) were conducted for interval scaled variables (total scores of the four dimensions of acceptance and usability) and Kruskal–Wallis H test for ordinal scaled variables (overall evaluation). P < 0.05 was considered to be significant for all analyses. The significance level was not adjusted as the tests served to generate hypotheses.

Qualitative data analysis

Qualitative data analysis was used to analyze the open field question using an inductive approach. Responses were categorized into five main categories: (1) negative appraisals, (2) positive appraisals, (3) suggestions for improvement, (4) not related to the website (5) no comment. Responses that included a number of themes were subdivided into various units and separately categorized. The coding was carried out by three members of the research team (LT, JT, SL).

Results

During the investigation period of 24 months, 14.000 to 36.000 visitors per month were registered through web analysis software. 1,030 visitors of the portal started the web-based user survey and 819 gave consent to participate at the beginning of the questionnaire. Of these, 314 completed the questionnaire (38.3% of those who agreed to participate). Finally, 252 participants gave their consent for the use of data at the end of the questionnaire, indicating that they answered the questionnaire in a meaningful way (e.g., did not answer the questions simply to have a look at the questionnaire) and that their data can be used for statistical analyses (see Fig. 1).

Figure 1 Participant flow chart.

Participants

Of the 252 respondents, 55.2% (n = 139) were affected from mental disorders. The respondents were predominantly female (64.3%, n = 162), well-educated (middle or high educational level: 75.8%, n = 191) and had a mean age of 42.2 years (SD = 15.0). The majority of the participants (90.5%, n = 228) are using the internet (almost) every day. 57.5% of respondents (n = 145) stated that they learned about the portal through online search for mental illnesses. 14.3% (n = 36) learned about the portal through the projects’ media campaign (cinema adverts, poster, YouTube channel, postcards). Of the total sample, 73.4% (n = 185) reported that they were visiting the portal for the first time. For detailed baseline characteristics and frequency distributions of access paths and website use see Table 1.

Table 1 Descriptive characteristics and frequency distributions of access paths and website use (N = 252).

Variables		n	%	
Gender	Female	162	64.3	
Age (M = 42.2, SD = 15.0)	≤45	135	53.6	
	>45	117	46.4	
Education	Low	61	24.2	
	Middle	66	26.2	
	High	125	49.6	
Experience with mental disorders	Affected people	139	55.2	
	Relatives	65	25.8	
	Experts	25	9.9	
	None	23	9.1	
Residential area	Region of Hamburg	62	24.6	
	Other regions	190	75.4	
Internet usage	(Almost) every day	228	90.5	
	At least once a week	21	8.3	
	At least once a month	2	0.8	
Access to the portal	Directly	122	48.4	
	Via search engine	102	40.5	
	Via referring website	28	11.1	
Awareness of the portal through	Online searches for mental illnesses	145	57.5	
	Personal recommendation	27	10.7	
	Newspaper article	19	7.5	
	Cinema advert	19	7.5	
	Poster	7	2.8	
	YouTube	6	2.4	
	Postcard	4	1.6	
	Other	60	23.8	
Frequency of use	First time	185	73.4	
	<5 times	42	16.7	
	>5 times	12	4.8	
	>10 times	13	5.2	
Date of attendance	Before the integration of the first PtDA	156	61.9	
	After the integration of the first PtDA	96	38.1	

Acceptance of the portal

Table 2 shows the percentage of users who agreed/disagreed to statements covering several aspects of acceptance, ordered separately for each dimension by the percentage of participants who agreed.

Table 2 User ratings on perceived ease of use, perceived usefulness, attitude towards using the portal, and perceived trust (N = 252).

Variables	Agree % (n)	Somewhat agree % (n)	Somewhat disagree % (n)	Disagree % (n)	
Perceived ease of use	
The font of the website is easy to read	71.0 (179)	25.0 (63)	1.6 (4)	2.4 (6)	
The website is easy to use	58.7 (148)	33.7 (85)	5.2 (13)	2.4 (6)	
The presentation of the information is clearly arranged	52.0 (131)	39.7 (100)	4.4 (11)	4.0 (10)	
The design of the website is appealing	52.8 (133)	39.3 (99)	4.8 (12)	3.2 (8)	
The information on the website is easy to understand	63.5 (160)	30.2 (76)	3.2 (8)	3.2 (8)	
The colors of the website are pleasant	54.0 (136)	38.5 (97)	5.6 (14)	2.0 (5)	
The pictures on the website are appropriate	44.0 (111)	46.8 (118)	6.3 (16)	2.8 (7)	
I can quickly find the information that is important to me	48.0 (121)	40.5 (102)	6.3 (16)	5.2 (13)	
Perceived usefulness	
The content of the website is interesting	61.1 (154)	31.7 (80)	4.4 (11)	2.8 (7)	
All in all, the website is useful for me	48.8 (123)	40.1 (101)	8.3 (21)	2.8 (7)	
The amount of information presented on the website is appropriate	44.8 (113)	43.3 (109)	8.7 (22)	3.2 (8)	
The website contains information that I need	47.2 (119)	40.1 (101)	9.5 (24)	3.2 (8)	
The information on the website has helped me with my concerns	40.1 (101)	42.9 (108)	12.7 (32)	4.4 (11)	
Through the website, I received references to other sources	39.7 (100)	44.8 (113)	11.9 (30)	3.6 (9)	
By using this website I have learned something new	37.3 (94)	41.3 (104)	15.5 (39)	6.0 (15)	
Now I’m able to talk better about mental disorders with my relative being affecteda	21.5 (14)	50.8 (33)	21.5 (14)	6.2 (4)	
Now I’m able to talk better about mental disorders with my health professionalb	20.9 (29)	38.8 (54)	19.4 (27)	20.9 (29)	
Now I’m able to talk better about mental disorders with my relativeb	23.7 (33)	27.3 (38)	26.6 (37)	22.3 (31)	
Attitude towards using	
I would recommend the website to others	55.6 (140)	34.5 (87)	6.0 (15)	4.0 (10)	
I will revisit the website if needed	64.3 (162)	28.6 (72)	4.4 (11)	2.8 (7)	
Perceived trust	
The information on the website is trustworthy	59.1 (149)	36.9 (93)	2.0 (5)	2.0 (5)	
The information on the website is up to date	48.0 (121)	45.6 (115)	4.0 (10)	2.4 (6)	
Notes.

a Sample size was reduced to n = 65 respondents that reported being relative of a person with mental disorders.

b Sample size was reduced to n = 139 respondents that reported being affected by a mental disorder

Perceived ease of use

A total of 89 to 96% of participants agreed with the particular statements concerning the perceived ease of use.

ANOVAs yielded no significant main effects of participants’ characteristics for items associated with perceived ease of use (see Table 3).

Table 3 Effects of different participants’ characteristics on the perceived ease of use (N = 252).

		N	M	SD	F	p	
Sex	Female	162	3.49	0.51	3.35	0.068	
	Male	90	3.36	0.63			
Age	≤45	135	3.46	0.56	0.30	0.586	
	>45	117	3.43	0.56			
Educational level	Low	61	3.33	0.68	2.22	0.111	
	Middle	66	3.53	0.39			
	High	125	3.46	0.57			
Residential area	Region of Hamburg	62	3.42	0.56	0.18	0.668	
	Other regions	190	3.45	0.56			
Experience with mental disorders	Affected people	139	3.41	0.58	1.15	0.328	
	Relatives	65	3.44	0.56			
	Experts	25	3.61	0.44			
	None	23	3.54	0.54			
Frequency of use	First time users	185	3.43	0.59	0.63	0.428	
	Multiple users	67	3.49	0.46			
Date of attendance	Before the integration of the first PtDA	156	3.40	0.60	2.99	0.085	
	After the integration of the first PtDA	96	3.52	0.48			
Notes.

M mean

SD standard deviation

Table 4 Effects of different participants’ characteristics on the perceived usefulness (N = 252).

		N	M	SD	F	p	
Sex	Female	162	3.49	0.51	3.35	0.068	
	Male	90	3.36	0.63			
Age	≤45	135	3.46	0.56	0.30	0.586	
	>45	117	3.43	0.56			
Educational level	Low	61	3.33	0.68	2.22	0.111	
	Middle	66	3.53	0.39			
	High	125	3.46	0.57			
Residential area	Region of Hamburg	62	3.42	0.56	0.18	0.668	
	Other regions	190	3.45	0.56			
Experience with mental disorders	Affected people	139	3.41	0.58	1.15	0.328	
	Relatives	65	3.44	0.56			
	Experts	25	3.61	0.44			
	None	23	3.54	0.54			
Frequency of use	First time users	185	3.43	0.59	0.63	0.428	
	Multiple users	67	3.49	0.46			
Date of attendance	Before the integration of the first PtDA	156	3.40	0.60	2.99	0.085	
	After the integration of the first PtDA	96	3.52	0.48			
Notes.

M mean

SD standard deviation

Perceived usefulness

Concerning the perceived usefulness, the items concerning the usefulness of the content (interesting, new, appropriate amount of information, helpful, useful) gained the highest level of approval (79–93%). Lower levels of agreement from the perspective of the respondents living with mental disorders were shown for statements concerning the improvement of the communication with relatives or health care providers (51 respectively 60%). Concerning the affected peoples’ relatives, 72% confirmed that they were now able to talk better about mental disorders with their relative being affected.

ANOVAs yielded no significant main effects of participants’ characteristics for items associated with perceived usefulness (see Table 4).

Attitude towards using the website

Concerning the attitude towards using the website, more than 90% of the respondents agreed that they would recommend the website to others respectively would revisit the website if needed.

ANOVAs yielded no significant main effects of participants’ characteristics for items associated with the attitude towards using the website (see Table 5).

Table 5 Effects of different participants’ characteristics on the attitude towards using the website (N = 252).

		N	M	SD	F	p	
Sex	Female	162	3.50	0.64	0.36	0.549	
	Male	90	3.44	0.81			
Age	≤45	135	3.46	0.72	0.36	0.551	
	>45	117	3.51	0.69			
Educational level	Low	61	3.42	0.86	0.31	0.732	
	Middle	66	3.50	0.61			
	High	125	3.50	0.67			
Residential area	Region of Hamburg	62	3.40	0.75	0.99	0.322	
	Other regions	190	3.51	0.69			
Experience with mental disorders	Affected people	139	3.46	0.74	1.88	0.133	
	Relatives	65	3.45	0.67			
	Experts	25	3.78	0.38			
	None	23	3.35	0.76			
Frequency of use	First time users	185	3.44	0.74	1.93	0.166	
	Multiple users	67	3.58	0.57			
Date of attendance	Before the integration of the first PtDA	156	3.48	0.71	0.01	0.941	
	After the integration of the first PtDA	96	3.48	0.69			
Notes.

M mean

SD standard deviation

Perceived trust

The majority of respondents (94–96%) agreed that the information on the website was trustworthy and that the information on the website was up to date.

For items associated with perceived trust, ANOVAs yielded no significant main effects of participants’ characteristics (see Table 6).

Table 6 Effects of different participants’ characteristics on the perceived trust (N = 252).

		N	M	SD	F	p	
Sex	Female	162	3.48	0.58	0.33	0.565	
	Male	90	3.43	0.61			
Age	≤45	135	3.49	0.60	0.58	0.446	
	>45	117	3.43	0.59			
Educational level	Low	61	3.39	0.71	0.77	0.466	
	Middle	66	3.46	0.49			
	High	125	3.50	0.58			
Residential area	Region of Hamburg	62	3.47	0.64	0.01	0.934	
	Other regions	190	3.46	0.58			
Experience with mental disorders	Affected people	139	3.45	0.61	0.50	0.685	
	Relatives	65	3.45	0.59			
	Experts	25	3.60	0.50			
	None	23	3.46	0.60			
Frequency of use	First time users	185	3.44	0.61	0.72	0.398	
	Multiple users	67	3.51	0.55			
Date of attendance	Before the integration of the first PtDA	156	3.42	0.64	2.44	0.120	
	After the integration of the first PtDA	96	3.54	0.50			
Notes.

M mean

SD standard deviation

Overall evaluation

Almost three thirds of the respondents (71.4%, n = 180) assessed the website as “very good” (n = 60) or “good” (n = 120). 21.0% of the participants (n = 53) rated the website as “satisfactory” (n = 39) or “sufficient” (n = 14). Only 7.5% (n = 19) marked the website as “deficient” (n = 13) or “insufficient” (n = 6). Overall, the ratings of the whole sample displayed a median of 2.0 (IQR = 1 − 6).

Kruskal–Wallis H test revealed no significant effect of age, educational level, place of residence, frequency of portal use and the date of participation (before/after the integration of the first PtDA) on overall evaluation.

Women were more satisfied with the portal than men (p = 0.019). The experience with mental disorders was also significantly associated with the overall evaluation (p = 0.037) with the best rating in experts and the worst rating in people without experiences with mental disorders.

Qualitative analysis

The open field question was responded by 58 participants. The answers were subdivided into 64 different statements. Seven participants explicitly mentioned that they had no comment. Five statements addressed the online survey and one statement addressed the general attitude towards people with mental disorders. All other statements refer directly to the e-health portal. There were 31 suggestions for improvement (e.g., the need for additional tools or topics, more in-depth information or regional expansion). Fifteen positive appraisals addressed knowledge and empowerment, the appropriate depth of information and the usefulness for newly diagnosed people among other topics. There were five negative appraisals concerning, for example, incomprehensible information (too many technical terms) or the insufficient suitability for adults with bipolar disorders.

Discussion

As a consequence of multiple barriers in mental health service provision and access, a considerable proportion of persons living with mental disorders do not receive adequate treatment (Wang et al., 2007). Internationally, but not yet in Germany, mental health services have increasingly expanded into online environments leading to the development of e-mental health services. Within the framework of an intersectoral research network the e-health portal www.psychenet.de addressed at individuals with mental disorders, their relatives and service providers has been developed recently. In this online study, acceptance regarding design and content of the portal was investigated.

In the present study, 252 users of the e-mental health portal www.psychenet.de were included. Overall, the portal was assessed as “good” or “very good” by a substantial percentage of respondents. Moreover, high degrees of approval were found for statements on perceived ease of use. Comparable rates of agreement were found in an evaluation study on the usability of a web-based patient information system for individuals with severe mental health problems (Kuosmanen et al., 2010). Likewise, high levels of agreement were shown for statements on the behavioral intention to use the portal or to recommend it to others and regarding the trustworthiness of the portal. Lower levels of agreement were partly shown for some statements on the perceived usefulness. Concerning the usefulness of the portal in improving communication, relatives show higher levels of agreement than the respondents living with mental disorders. In a recent study, Berk et al. (2013) reported comparatively higher levels of agreement regarding the usefulness of a website containing guidelines for caregivers of adults with bipolar disorder. Likewise, a study on the acceptance of a web-based e-health intervention for parents of children with infantile hemangiomas showed higher agreement rates (De Graaf et al., 2013). It is assumed that the higher acceptance was due to the fact that the respective website was aimed at one target group (caregivers) and one narrowly defined topic (bipolar disorders, infantile hemangiomas). However, in an evaluation study on the user acceptance of a website for cancer patients with a more broad range of topics, higher levels of agreements were reported for ease of use compared to usefulness as it was also shown for the current study (Wallwiener et al., 2010). Additionally, it should be noted that such comparisons are difficult to interpret as the studies probably varied substantially with respect to relevant characteristics such as ways of recruitment, response rates and users’ experience with the respective portal.

Facing the fact that there were no effects of different participants’ characteristics on the perceived ease of use, the perceived usefulness, the attitude towards using the website and the perceived trust, it can be assumed that the e-mental health portal is suitable for a broad range of users. Concerning the overall evaluation, there are some differences depending on users’ characteristics: women are more satisfied with the portal than men. As there are no sex differences regarding the other items, the difference results maybe from a differing answering behavior regarding overall ratings. Additionally, the overall evaluation depends on the experience with mental disorders, indicating that experts are more pleased with the portal than affected people and relatives and all these three groups are more pleased than people without experiences with mental disorders. As the portal is targeted to experts, affected people and relatives, the last-mentioned result is not surprising.

Fortunately, the educational level had no influence on the acceptance and usability of the portal, suggesting that respondents with lower educational level are also able to benefit from the information presented at the portal. However, most respondents are well-educated and we do not know if this reflects the typical users’ characteristics or if well-educated users are more likely to participate in the survey.

This analysis of acceptance offers preliminary evidence that the e-mental health portal www.psychenet.de appears to be a usable, useful and trustworthy publically available information resource for adults living with mental illness, their relatives and experts working with mental disorders. The acceptance of the portal is further resembled by the high percentage of respondents that agreed their intention to recommend and to revisit the portal in case of necessity. The results of the web-analysis reported by Dirmaier et al. (2015) confirmed that the website is usable and highly accessed. Nonetheless, lower agreement levels concerning the usefulness of the portal on a behavioral level were observed. Thus, integrating content that supports active patient behavior regarding communication with relatives and with health care providers as provided by high quality patient decision aids (PtDAs) might improve the usefulness of the e-health portal. Previous analyses do not show an influence of the availability of the first decision aid on the acceptance of the portal. However, three of the four PtDAs were only available during the last weeks of the survey period. In order to further improve acceptance of the portal by targeting the offers of the portal to the users’ needs, qualitative studies are requested to identify topics that are of high relevance to the users but have not been addressed until now.

Limitations

Due to methodological limitations the results of the study need to be interpreted with caution. First of all, convenience sampling was used by informing users about the survey without attracting attention and not actively recruiting. This resulted in a relatively small number of respondents—compared with the number of website users and the number of people who started the survey. It is assumed, that respondents might have had an incentive to participate in the study as a consequence of being either particularly satisfied or dissatisfied with the offers presented at the portal. However, the positive ratings of the respondents suggest that they might have been motivated rather by their satisfaction than dissatisfaction with the system. However, high attrition rates are a common problem in online-surveys (Thielsch & Weltzin, 2012).

As we do not know if the investigated sample was representative (e.g., if the completers represent the typical users’ characteristics) the results presented here might overestimate the acceptance of the portal. Most non-completers (46% of the people who started the questionnaire) discontinued the questionnaire during the first page (introduction), another 9% of the non-completers discontinued on the second page (informed consent to participate). When it came to the questions concerning the acceptance and the usability of the portal, further 11% discontinued, maybe due to the complex matrix character of these questions. The other non-completers discontinued at other pages. Future evaluations should be conducted using probability sampling methods to confirm the present findings. As we used hardly any standardised instrument, the comparability of our results is limited. However, the questionnaire was developed based on widespread theories and evidence on acceptance of information technologies (Chau & Hu, 2002; Davis, 1989; Kerr et al., 2006; Lemire et al., 2008; Wu et al., 2008) and pilot tested among 10 participants. Furthermore, in order to provoke a definitive choice, no mid-point was provided. Due to the forced choice, the use of a 4-point scale might have led to a biased rating. However, Weijters, Cabooter & Schillewaert (2010) assumed that ambivalent or neutral respondents tend to rate negatively in the absence of a midpoint.

Moreover, as we used self-reported information on the respondents’ experience with mental disorders, the validity of this information is limited.

Conclusions

Despite the methodological limitations, this study provides first evidence on the acceptance of the e-mental health portal www.psychenet.de. The results on the usefulness of the portal showed that there is still room for improvement. It is assumed that the portal empowers people with mental disorders and their relatives by facilitating to gather high-quality evidenced-based information about their illness, to rapidly find the right treatment services without great effort, and to prepare for health care provider contacts. Within the framework of this project, PtDAs for common mental disorders (i.e., depression, anxiety disorders, psychosis) supporting active user behavior were developed and implemented on the e-health portal www.psychenet.de based on a comprehensive mixed-methods needs assessment study. In addition to the PtDAs, self-management tools are currently being evaluated.

Supplemental Information

Data S1 Raw data

Click here for additional data file.

psychenet is a project network supported by the German Federal Ministry of Education and Research in the region of Hamburg which consists of more than 100 scientific and medical institutions, counselling centers, the Senate and the Chamber of Commerce of the Free and Hanseatic City of Hamburg, companies, as well as patients’ and relatives’ associations (2011–2015). The vision of the project is to promote mental health today and in the future, concerning early diagnosis and effective treatment of mental illnesses. For more information and a list of all partners, please visit www.psychenet.de.

Additional Information and Declarations

Competing Interests

Author Contributions

Human Ethics

Data Availability

All authors are (or were) members of psychenet, subproject II: communication platform and interactive internet portal. It is a public-funded network focusing research on mental disorders, the authors do not earn money with the web portal and do therefore don’t have direct financial interests.

Lisa Tlach, Juliane Thiel and Sarah Liebherz conceived and designed the experiments, performed the experiments, analyzed the data, contributed reagents/materials/analysis tools, wrote the paper, prepared figures and/or tables, reviewed drafts of the paper.

Martin Härter and Jörg Dirmaier conceived and designed the experiments, performed the experiments, contributed reagents/materials/analysis tools, wrote the paper, reviewed drafts of the paper.

The following information was supplied relating to ethical approvals (i.e., approving body and any reference numbers):

Ethics committee of the Hamburg Medical Association (Process number: PV4157).

The following information was supplied regarding data availability:

The raw data has been supplied as Data S1.

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
