# Peer review of "Acceptance of the German e-mental health portal www.psychenet.de: an online survey"

_PeerJ, doi:10.7717/peerj.2093_

## Round 0.1 · original submission · Major Revisions

Editor’s Response
Manuscript no: #1970
Title: Acceptance of the German e-mental health portal

Comments for the author:

• Overall, the paper requires more in-depth analyses and discussion regarding the question of ‘what aspects of the website are useful’ and ‘for whom’? As it stands, the paper only very broadly looks at whether the website as a whole is useful for those who access it.
• The demographic data could be used in more detail to analyse who the website was useful for (not just education differences) – what about first time vs. multiple users, or gender and age differences, those in North Germany vs. other regions? Also, the authors mention that the website content changed during the middle of the survey period to include PtDAs – the authors should use the survey date data to examine if the website became more useful after the introduction of these additional interactive elements.
• The poor sample representation is a significant problem. While this is mentioned in the discussion, is there anything more that could be done by the authors to justify the usefulness of the data obtained? Is there anything in the non-completers data to indicate why they did not complete the survey? Is there a particular question stage at which people tended to discontinue participation?
• The measures section is not described clearly enough and there are inconsistencies in the number of total items assessed (e.g. initially states 29 items, but this doesn’t seem to add up in Tables 1 and 2). Please state each measure used in the survey clearly.
• The whole paper needs an edit for language and grammar clarity. There are many instances where the grammar is incorrect.

·

Basic reporting

The authors describe the acceptance of a recently developed e-mental health portal and had a special interest in the effect of education level on acceptance of the portal. The introduction is accurate and well written. The structure of the manuscript conforms to PeerJ standard. Figure 1 is relevant but could be labelled more specifically and raw data is supplied by the authors.

Experimental design

The research question is well defined, but no hypotheses were included. Although the authors did not make use of standardised questionnaires, they have based their questionnaires on relevant theory. Methods are described in sufficient detail, but the description of the questionnaire items was slightly confusing to me. The section ‘data collection’ starts out by stating that 29 items were developed for the study. The next section describes the baseline characteristics items, which adds up to 9 in total (which corresponds with table 1). Then, the acceptance and usability items are described, which add up to 21 instead of 19 as stated in the first sentence of this paragraph. Furthermore, this number does not match up with table 2, as there are 22 items in total in the table. The differences seem to come from the ‘perceived ease of use’ questions (10 items according to text, 8 items in table) and ‘perceived usefulness (7 items according to text, 10 items in table). Finally, there are 2 items addressing the overall evaluation, which would bring the total to 33 (instead of 29).

Validity of the findings

The authors had an interest in the effect of education level on acceptance of the portal. As they state in the ‘participants’ section, the sample was generally well educated, so I was wondering if the fact that a relatively small portion of the sample had low education may have influenced their findings? The authors do not address this in the limitations, but it would be worth a mention.
Furthermore, I was wondering at what point during the visit to the portal people were asked to complete the survey. The majority of participants completed the survey during their first visit and I was wondering how much of the portal they would have looked at by the time they completed the survey. Do the authors have any data on how many pages participants viewed, for example? Moreover, some of the items seem to imply that participants have spent some time looking at the portal.

Additional comments

While reading the first paragraph of the results, I was wondering how many people visited the portal in the 24 month period that the survey ran to get an idea of uptake of the survey. I noticed later that this is mentioned in the discussion, but I would suggest moving that to the results section and not present new information in the discussion.

In figure 1, the last step mentions exclusion of “unplausible data”. This is not mentioned in the method or results section, so it is unclear what is meant by this. Furthermore, no data seems to have been excluded (n is still 252), so perhaps this can be omitted altogether?

The summary of results in the discussion can in my view be condensed. It would be sufficient to summarise the main results and there is generally no need to repeat specific percentages and items in the discussion. Also, I think that the conclusion should be kept to one paragraph (the second paragraph would fit better earlier in the discussion).

·

Basic reporting

This paper describes an evaluation of a German website/portal for mental health resources. It is generally a well-written and structured paper.

Experimental design

Why did you not use questionnaires of which some psychometric data are known, such as the System Usability Questionnaire?

How were mental disorders ascertained and what is known about the validity and reliability of these instruments?

I did not see a conflict of interest statement. Were the authors involved in the development of the website or any of its content?

Validity of the findings

A considerable amount (505) of people who started the questionnaire and consented to participate (819) did not finish the questionnaire. That is a loss of 62%. Do you have any idea why? How can this have affected the results? Are there any demographic data known of this group?

Additional comments

Generally, I have some doubts about the relevance of this paper. It is probably a useful reference to publish on the website, but this study is not conducted in such a way that other website developers can learn something.

---

## Round 0.2 · Minor Revisions

Editor’s Response. Manuscript no: #1970
Title: Acceptance of the German e-mental health portal

Comments for the author:

• The paper has been improved by the additional changes made by the authors.

• However, the new additional analyses involving testing the differences in acceptability/useability ratings are somewhat unclear. Can the authors confirm and include the following in the paper: I assume the ANOVAs involved creating total scale scores for each of the acceptability/useability domains by summing the individual relevant items (e.g. Ease of use, Perceived usefulness, Attitudes, Perceived trust), and then testing for differences in the means of these scale scores based on the characteristics of interest. E.g. Men scored a mean and SD of ??? on the total Perceived usefulness scale as opposed to women who scored a mean and SD of ???, and the ANOVA showed there were no statistical differences in the mean scores. I had wondered whether the differences were actually tested for each ‘item’ (rather than the total scale scores), but could not then see how the use of ANOVA would be appropriate given the individual items are really ordinal rather than continuous (e.g. agree, somewhat agree, somewhat disagree, disagree). Also, there seems to be limited usefulness in testing for differences each item rather than the broad domain. So, please clarify in the paper that total scale scores were calculated, and that the ANOVAs tested for differences in the means of these scores based on the characteristics of interest. Please include 4 supplementary tables (one for each domain) showing the results for these analyses (e.g. means and SDs for 4 scales (Ease of use, Perceived usefulness, Attitudes, Perceived trust), categorised by different sample characteristics, with p-values).

• Please include the information in your rebuttal letter about non-completers in the limitation section of the manuscript: Most non-completers (46% of the people who started the questionnaire) discontinued the questionnaire during the first page (introduction), another 9% of the non-completers discontinued on the second page (informed consent to participate). When it came to the questions concerning the acceptance and the usability of the portal, further 11% discontinued, maybe due to the complex matrix character of these questions. The other non-completers discontinued at other pages.

• Figure 1 still seems a bit strange. How did the sample get from 314 to 252? What was un-usable about the data of those who were excluded? The text says that “1030 visitors of the portal started the web-based user survey. Of these, 314 completed the questionnaire (38.3% of those who agreed to participate). Finally, 252 participants gave their consent for the use of data (see Figure 1).” This suggests that the first box in Figure 1 should be 1030, the second 314 and the third 252. The order of flow in the Figure, should reflect the temporal order in which participants dropped-out/were excluded.

---

## Round 0.3 · accepted · Accept

Thank you for the last round of Minor Revisions. Congratulations, the paper has now been accepted.